# Emerging Strategies in Proteolysis-Targeting Chimeras (PROTACs): Highlights from 2022

**DOI:** 10.3390/ijms24065190

**Published:** 2023-03-08

**Authors:** Rekha Tamatam, Dongyun Shin

**Affiliations:** 1College of Pharmacy, Gachon University, 191 Hambakmoe-ro, Yeonsu-gu, Incheon 21936, Republic of Korea; 2Gachon Pharmaceutical Research Institute, Gachon University, 191 Hambakmoe-ro, Yeonsu-gu, Incheon 21936, Republic of Korea

**Keywords:** targeted protein degradation, proteolysis-targeting chimeras, selectivity, permeability, linker flexibility

## Abstract

Targeted protein degradation (TPD) is a promising therapeutic modality that has garnered attention in academic, industrial, and pharmaceutical research for treating diseases such as cancer, neurodegenerative disorders, inflammation, and viral infections. In this context, proteolysis-targeting chimeras (PROTACs) present a reliable technology for degrading disease-causing proteins. PROTACs complement small-molecule inhibitors, which primarily rely on direct protein regulation. From concept-to-clinic, PROTACs have evolved from cell impermeable peptide molecules to orally bioavailable drugs. Despite their potential in medicinal chemistry, certain aspects regarding PROTACs remain unclear. The clinical significance of PROTACs is primarily limited owing to their lack of selectivity and drug-like properties. This review focused on recently reported PROTAC strategies, particularly in 2022. It aimed to address and overcome the challenges posed by classical PROTACs by correlating them with emerging approaches with improved selectivity and controllability, cell permeability, linker flexibility, druggability, and PROTAC-based approaches, developed in 2022. Furthermore, recently reported PROTAC-based approaches are discussed, highlighting each of their advantages and limitations. We predict that several improved PROTAC molecules will be accessible for treating patients exhibiting various conditions, including cancer, neurodegenerative disorders, inflammation, and viral infections.

## 1. Introduction

Classic drug discovery enables the identification of small-molecule compounds that inhibit target proteins by occupying well-defined active or allosteric sites in receptors, enzymes, or ion channels [1,2,3]. However, the lack of binding sites in various proteins renders them undruggable [4]. Approximately 80% of proteins, such as transcription factors and non-enzymatic and scaffolding proteins, are considered undruggable targets [5,6]. Alternative modalities have eventually emerged to address this issue [7]. Among these, proteolysis-targeting chimeras (PROTACs) caused a paradigm-shift in drug discovery with its ability to harness a natural degradation pathway (ubiquitin–protease system) for targeted protein degradation (TPD) [8]. From concept-to-clinic, PROTACs have evolved from being cell impermeable peptide molecules to orally bioavailable drugs. PROTACs are heterobifunctional molecules comprising ligands of the protein of interest (POI) and E3 ligase, connected by a linker. These heterobifunctional degraders bind with a POI on one end and ubiquitin E3 ligase on the other end, connected by a linker to form a ternary complex, further activating the ubiquitin–proteosome system to degrade the target protein (POI) (Figure 1).

A single PROTAC molecule can trigger multiple POI molecules and therefore act catalytically through an event-driven mechanism. PROTACs have garnered considerable attention as they target the “undruggable” [9] targets, thus overcoming drug resistance [10]. Clinical trials have been initiated for more than 10 PROTAC molecules [11]. Although remarkable progress has been made with respect to these chimeras, certain inherent shortcomings are still associated with them: (a) they are less selective and hence degrade other proteins (off-target toxicity) and exhibit non-selective degradation in an undesired tissue (on-target effect); (b) they induce uncontrollable catalytic degradation; (c) their chemical composition lies beyond the Lipinski’s rule of five, thus impeding cell permeability; (d) linker flexibility is essential; and (e) some of the small-molecule degraders exhibit difficulty in targeting hydrophobic pockets owing to the lack of well-defined binding sites [12]. However, these limitations were addressed by a prior novel PROTACs report [13]. The present review is primarily focused on collating information that addresses the above-mentioned challenges associated with classical PROTACs and discusses the strategies developed regarding the same in 2022. Furthermore, the advantages and limitations of newly emerging TPD strategies were highlighted. Specifically, the review is categorized as modalities with the following improvements:Selectivity and controllability;Cell permeability;Linker flexibility;Druggability;PROTAC-based approaches.

## 2. Discussion

### 2.1. Modalities with Improved Selectivity and Controllability

Classical small-molecule PROTACs exhibit unfavorable pharmacokinetics and hence lack cell or tissue selectivity, which might cause systemic toxicity owing to nonspecific distribution in normal or undesired tissues [14]. Ligand-modification strategies such as folate–, antibody–, and aptamer–PROTAC conjugates have been developed recently for tumor-targeted delivery [15,16,17]. These ligand–PROTAC conjugates exhibited advantages such as enhanced accumulation in tumors and antitumor potency in vivo and increased cellular uptake. Despite the superior catalytic behavior exhibited by PROTACs, their off-target effect remains a challenge to be addressed. Off-target effects can be reduced by regulating PROTAC activity, thereby improving target specificity and antitumor activity [18]. This can be achieved by spatiotemporally controlling the biological system with light. In the presence of light, PROTAC activity was controlled by utilizing photoactive ligands, viz., cages and photo switches. The caging approach is based on rendering the PROTACs inactive with a photocleavable group that blocks the binding process with either the E3 ligase or POI. Uncaging with light at an appropriate wavelength resulted in the generation of active PROTACs, leading to TPD. These heterobifunctional degraders were also activated in response to internal or external stimuli and called smart activable PROTACs. Based on their activation mechanism, reduction-, enzyme-, and hypoxia-activable PROTACs have been developed. However, limited tumor penetration, low serum stability, and heterogenous expression of receptors in various tumor cells were found to be their shortcomings. In this regard, several new strategies were developed to overcome cell or tissue selectivity, tumor specificity, and controllability, as discussed below (Figure 2).

#### 2.1.1. Aptamer-Based PROTAC

Single-stranded DNA or RNA oligonucleotides <100 nucleotides in length are called aptamers [19]. They exhibit properties such as high affinity, high specificity, low toxicity, enhanced tissue penetration, easy chemical synthesis, and excellent stability. Nucleolin is regarded as an important target in cancer therapeutics; it is a multifunctional nucleolar protein localized in the nucleoplasm, cytoplasm, and cell membrane that plays a vital role in performing a wide array of cellular activities [20]. In general, PROTACs are hydrophobic and display low cell-type selectivity. To address this, Tan et al. (2022) reported the first proof-of-concept example by constructing a novel PROTAC-utilizing nucleic acid aptamer, a 26-base guanine-rich ssDNA, as the targeting ligand of nucleolin [21]. The enhanced expression of nucleolin on the cell surface is generally restricted in tumor cells. Therefore, the aptamer was internalized into the tumor cells through nucleolin-dependent macropinocytosis, thus acting as a potential tumor-targeting element and anticancer therapeutic agent [22]. The PROTAC **1** was designed by conjugating an aptamer with a small molecule, VHL, which recruited an E3 ligase ligand via a DBCO-azide click reaction connected by PEG linker (Figure 3). The resultant aptamer-constructed PROTAC induced nucleolin degradation in breast cancer cells, both in vitro and in vivo. Furthermore, this approach demonstrated selective degradation of tumor cell proteins, good water solubility, and excellent serum stability. This is a promising strategy, with its only limitation being the shortage of aptamers that specifically bind to E3 ligases. In contrast to the recently reported aptamer conjugate BET–PROTAC, a targeted delivery tool for specific cell types [23], this novel PROTAC compound promoted the generation of nucleolin–PROTAC–VHL ternary complex by using a nucleolin-binding aptamer as a ligand.

Aptamers are recognized artificially via an iterative in vitro selection strategy called the systematic evolution of ligands by exponential enrichment (SELEX) [24,25,26]. Modification of microenvironmental pH, complementary stands, or metabolic molecules leads to switching of the unique structure of aptamers, allowing for the selective modulation of aptamers and POI interactions [27,28,29]. Recently, Xie et al. (2023) developed apt-PROTAC **2** in which the aptamer directed the oncogenic nucleolin to E3 ligase for degradation [30]. This PROTAC **2** was synthesized via the click reaction between thalidomide-*O*-amido-propargyl, a cereblon ligand (CRBNL), and azide-modified AS1411 aptamer. A series of thymidines or PEG groups were employed as the linkers. Furthermore, a light-inducible nucleolin degrader, termed opto-PROTAC, was developed by hybridizing the apt-PROTAC with the complimentary photolabile oligonucleotide (CP). Upon irradiation with UV (365 nm), light-triggered breakage of photolabile bonds resulted in the liberation of the active PROTAC degrader (Figure 4). The major advantages of this strategy include (a) apt-PROTACs act as an alternative approach for undruggable TPD owing to their ability to recognize all types of proteins, specifically mutants; (b) it is a convenient mode of synthesis by utilizing nucleotides as linkers; (c) it provides convenient spatiotemporal regulation of the activity of apt-PROTACs by adopting conformational switchable aptamers; and (d) the selective accumulation of apt-PROTACs in cancer cells resulting in reduced on-target toxicity in normal cells.

#### 2.1.2. Pre-PROTACs

To improve therapeutic efficacy and reduce systemic toxicity, drugs are designed to be activated by specific triggers in a predetermined pathway. Reactive oxygen species (ROS), such as hydrogen peroxide, superoxide, and hydroxyl radicals, play an important role in regulating signaling pathways [31]. Low ROS levels are maintained in normal cells, while it is much higher in the tumor microenvironment [32]. Chen et al. (2022) designed ROS-responsive Pre-PROTACs **3** by conjugating an ROS-triggered leaving group, arylboronic acid, to the parent PROTACs [33]. This PROTAC molecule **3** was designed by recruiting a CRBN ligand and BRD3 protein ligand to opposite ends connected by varying linkers (Figure 5). These PROTACs showed excellent TPD of BRD3. Its major limitations are as follows: (a) the occasional generation of unwanted by-products, although the caging strategy is efficient in TPD and (b) the inhibitory activity of Pre-PROTAC is sensitive to the cell line with high ROS levels. Notably, this is the first report of ROS-activated Pre-PROTAC, which was proved to have potential in controlling PROTAC activation through an endogenous ROS-related microenvironment.

#### 2.1.3. PS-Degrons

He et al. (2022) proposed PS-Degrons involving the activation of a TPD platform by adopting a light-triggered pathway [34]. The PS-Degrons **4** were designed by conjugating photosensitizers and target protein ligands (Figure 6). This methodology resulted in the controlled knockdown of the target protein, human estrogen receptor α, with high spatiotemporal precision via light-triggered ^1^O_2_. Furthermore, PS-Degrons enable elevated anti-proliferation in MCF7 cells. This modality complements existing technologies by overcoming systemic toxicity.

#### 2.1.4. AP-PROTACs

Despite the rapid progress achieved by PROTACs, the spatiotemporal control of degradation still remained elusive. Methods to attain conditional degradation involved selective targeting or light-driven photo-pharmacology [35]. Traditional examples of light-driven photo-pharmacology include caged and photo-switchable PROTACs, with irreversible and reversible modes of action, respectively. PHOtochemically Targeting Chimeras (PHOTACs) are a small group of photo-switchable PROTACs having an azobenzene photo-switch with an active E (trans) or Z (cis) isomer that induces TPD. Tate et al. (2022) reported a novel arylazopyrazole photo-switch (AP-PROTACs), **5** and **6** the first example of a non-azobenzene PHOTAC technology [36]. Using this, two types of AP-PROTACs were synthesized: (a) AP-PROTACs **5** with pan-bromodomain inhibitor JQ1 as the POI ligand and thalidomide as a CRBN recruiter and (b) AP-PROTACs **6** with a multi-kinase inhibitor as the PROTAC warhead, with lenalidomide as a CRBN-recruiting ligand connected by arylazopyrazole (AAP) linkers (Figure 7). These AP-PROTACs **5** and **6** have the following advantages: (a) AAP linkers acting as a potential “plug and play” photo-switch to control biological outcomes; (b) good PSS isomer abundance; (c) long z-isomer half-life; and (d) stable reversible switching for multiple cycles. Notably, to the best of our knowledge, this is the first example of a multi-target PROTAC that adopts a photo-switchable degradation pathway.

#### 2.1.5. RT-PROTACs

To overcome the limitation of systemic toxicity caused by undesirable off-TPD posed by classical PROTACs, Li et al. (2023) developed a novel radiotherapy-triggered PROTAC **7** (RT-PROTAC) prodrug strategy for spatiotemporally-controlled TPD through X-ray radiation [37]. RT-PROTAC **7** attaches a bromodomain (BRD)-targeting PROTAC to an X-ray inducible phenyl azide cage (Figure 8). This prodrug has been activated by X-ray radiation, both in vitro and in vivo. Thus, activated PROTACs **8** target BRD4 and BRD2 with remarkable antitumor potency in the MCF-7 xenograft model.

#### 2.1.6. NTR-Based PROTACs

The most common characteristic of tumors is their state of low oxygen levels or hypoxia, resulting in uncontrollable metastasis, therapeutic resistance, and poor prognosis [38]. Hypoxia is generally accompanied by high levels of nitroreductase (NTR). The catalytic behavior of PROTACs occasionally results in off-tissue target degradation, leading to potential toxicity. Thus, Zhu et al. (2022) developed the first NTR-responsive PROTAC **9** comprising a VHL E3 ligase ligand (caged by an NTR activating group) and an epidermal growth factor target protein ligand connected by a polyethylene glycol (PEG) linker (Figure 9) [39]. This strategy allowed precise control of the PROTAC **10** for selective TPD with good antitumor efficacy.

#### 2.1.7. ENCTACs

Hypoxia also affects gene expression and influences epigenetic modifications [38]. Xing et al. (2022) developed enzyme-derived clicking PROTACs **11** (ENCTACs) for stimulating responsive protein degradation in the tumor microenvironment [40]. ENCTACs depend on nitroreductase (NTR) enzyme for selectively forming degraders at the hypoxic site and result in targeted epigenetic BRD4 degradation. The design of NTR-oriented ENCTACs **11** comprises orthogonal cross-linking between the uncaged cysteine residue pre-modified with E3 ligase cereblon (CRBN) recruiting ligand, pomalidomide, and JQ1 (2-cyanobenzothiazole (CBT)-preconjugated BRD4 ligand), connected by a PEG_2_-NH_2_ linker (Figure 10). ENCTACs are formed locally within tumors only through the hypoxia-induced activation of the NTR enzyme. Its advantages include (a) precise modulation of hypoxia signaling in living mice, zebrafish, and live cells with solid tumors and (b) controlled cross-linking without prior covalent conjugation, maximizing pharmacological performance due to the large molecular weight. However, limitations of this approach include a lack of selectivity between normoxic and hypoxic cells, possibly resulting in detrimental off-target effects and the potential effect of ENCTACs on the stability of other proteins in the cells.

#### 2.1.8. POLY-PROTACs

Yu et al. (2022) reported a novel polymeric PROTAC **13** (POLY-PROTAC) nanoplatform for the tumor-targeted degradation of the bromodomain and extra-terminal (BET) protein, BRD4 [41]. POLY-PROTAC **13** consists of a VHL ligand and BRD4 protein ligand connected by optimizable linkers (Figure 11). Its advantages include (a) activation through multiple stimuli (extracellular matrix metalloproteinase-2, intracellular acidity, and reductive conditions), resulting in improved tumor accumulation, enhanced protein degradation, and deep tumor penetration; (b) tumor-specific PROTAC delivery being achieved owing to an extracellular tumor acidity-triggered biorthogonal click reaction; and (c) its combination with photodynamic therapy (PDT), suppressing more than 95% of MDA-MB-231 tumor growth in triple-negative breast cancer.

#### 2.1.9. Smart Nano-PROTACs

An interesting modality for cancer-specific protein degradation has been reported by Kanyi et al. (2021) [42]. The smart nano-PROTACs **15** comprise a semiconducting polymer backbone linked to cyclooxygenase1/2 (COX-1/2)-targeting PROTAC peptide (CPP) via a cathepsin B (CatB)-cleavable segment (Figure 12). Through the ubiquitin–proteasome system, COX-1/2 is degraded by CPP activation catalyzed by the tumor-overexpressed CatB. The advantages of this SPN_COX_-mediated technology include enhancement of tumor immunogenicity, intra-tumoral degradation of COX-1/2, and reprogramming of the immunosuppressive tumor microenvironment, thus inhibiting tumor growth, metastasis, and recurrence.

### 2.2. Modalities with Improved Linker Flexibility

#### Split-PROTACs

An inherent challenge associated with the activity of traditional PROTACs is the high dependency on the length and nature of the linker connecting the target protein and E3 ligase ligands. To address this, Kodadek et al. (2022) created “split PROTACs” **16** with optimizable linker flexibility (Figure 13) [43]. This involved combining the modified E3 ubiquitin ligand and POI ligand with residues suitable for coupling when mixed together in a combinatorial fashion. The aldehyde- and alkoxyamine-functionalized fragments were examined using oxime ligation methodology. This strategy eased the task of synthesizing several PROTACs. In addition, the PROTAC molecule assembled efficiently upon premixing high concentrations of the components and then adding them to the cells. The only limitation of this protocol was that at relatively low concentrations, the POI ligand and E3 ligase ligand displayed inefficient coupling in situ.

### 2.3. Modalities with Improved Cell Permeability

The molecular weight of PROTACs is considerably higher than that of traditional small-molecule inhibitors, potentially resulting in pharmacokinetic hurdles to their cell permeability. An attempt to address this issue was made by developing in-cell click-formed compounds called CLIPTACs [44]. This novel strategy enabled the intracellular assembly of PROTAC molecules from small fragments via a click reaction; the resulting compounds exhibited enhanced cell permeability. However, the success of CLIPTACs is limited by their low reaction yields, resulting in leftover unreacted fragments that interfered with the formation of ternary complex, a key step in protein degradation.

#### 2.3.1. Hydrophobic Tags

Neurodegenerative disorders are localized to the central nervous system (CNS). Despite the potential chemical knockdown of neurodegenerative disorder-related aggregation-prone proteins by PROTACs, these degraders have poor CNS permeability. Ishikawa et al. (2022) documented the conversion of PROTAC degraders into hydrophobic tags **17** (HyTs) [45], which are heterobifunctional molecules composed of POI ligand linked to a hydrophobic degron, viz., diamantane, that mimics the exposed hydrophobic region of misfolded proteins and is recognized by heat shock protein 70 kDa (HSP70) or C-terminal of HSC70-interacting protein (CHIP) [46]. Hydrophobic tagging technology utilized HSP70, the protein quality control machinery for TPD, via the proteasomal pathway (Figure 14). Consequently, the developed HyTs displayed potentially decreased levels of mutant huntingtin, an aggregation-prone protein. Moreover, the blood–brain barrier penetration potential and brain-to-plasma concentration values were calculated. The findings successfully detected a brain-permeable HyT using immobilized artificial membrane (IAM) chromatography analysis. The major advantages of this conversion were as follows: (a) more potent than the parent PROTAC, (b) a decrease in the molecular weight and number of hydrogen bond donors and acceptors, and (c) improved drug-like properties. Furthermore, this was the first example of a small-molecule hydrophobic tag targeting an aggregation-prone protein.

#### 2.3.2. Stapled PROTACs

In contrast to traditional PROTACs, the recently developed linear peptide-based PROTACs are considered undruggable because of their proteolytic stability and poor cellular uptake. To address this limitation, Hu et al. (2022) developed stapled (SP-)PROTACs **18** by combining peptide stapling with PROTAC [47]. The SP-PROTAC **18** structure comprises an MDM2/MDMX-targeting moiety, viz., PMI-N8A peptide, PEG linker, and VHL E3 ligase recruiting hydroxyproline (Hyp)- and homoleucine (Hle)-containing hexapeptide (LA-Hyp-Y-Hle-P) (Figure 15). The synthesized degraders showed improved proteolytic stability, better cellular permeability, good pharmacokinetic profile, and similar binding to MDM2 and MDMX. Furthermore, SP-PROTACs inhibited tumor progression in orthotopic colorectal cancer xenograft models by the simultaneous atypical degradation of both MDM2 and MDMX and p53 activation. The main advantages of SP-PROTACs are (a) target protein specificity, (b) high binding potency with both the target protein and E3 ligase, (c) ease of synthesis, (d) potent in vitro and in vivo activities, and (e) efficient death of cancer cells. There are certain limitations associated with SP-PROTACs. First, the hydrocarbon-stapled peptide has poor water solubility. Second, they cause deficient TPD owing to the cleavage of peptide-based E3 ligase-recruiting ligands in the cellular environment.

### 2.4. Modalities with Improved Druggability

In drug discovery and development, small-molecule inhibitors primarily target either the active site or an allosteric site of an enzyme to inhibit the function of the POI (occupancy-driven mode) [48]. In contrast, PROTACs are event-driven therapeutics, wherein one degrader molecule efficiently triggers the degradation of many POI molecules. PROTACs have been widely investigated for degrading previously undruggable targets. The characteristics of such undruggable targets include the lack of well-defined binding sites, non-catalytic protein–protein interaction functional modes, and ambiguous 3D structures. PROTACs negate the need for an active site owing to their event-driven pharmacology. Recent emerging approaches for targeting undruggable targets include covalent drugs, protein-based drugs, peptide-based drugs, bifunctional molecules, and RNA therapeutics.

#### 2.4.1. Bridged PROTACs

Classical PROTACs were successful in targeting druggable and difficult-to-drug proteins. Recently, degraders such as transcription factor TF-PROTACs [49] and RNA-PROTACs [50] have also been developed to target undruggable TFs and RNA-binding proteins, respectively, with small-molecule binding sites. Jin et al. (2022) introduced a novel strategy, the bridged PROTAC **19**, which specifically targets undruggable proteins by utilizing a small-molecule binder from the binding fragment of the POI [51]. This novel compound **19** successfully degraded the undruggable protein cyclin D1 via a unique binding mode. The bridged PROTAC **19** is composed of VHL E3 ligase, CDK4/6 binder connected to VHL E3 ligase by a linker, CDK4/6 as the bridged protein, and cyclin D1 as the target protein (POI) (Figure 16). Compared with CDK4/6 degraders and inhibitors, this novel platform showed remarkable efficiency in suppressing cancer cell proliferation. Moreover, bridged PROTACs formed quaternary complexes, such as cyclin D1-CDK6-PROTAC-VHL, and provided a proof-of-concept for degrading undruggable proteins lacking small-molecule binders.

#### 2.4.2. TF-DUBTACs

The targeted degradation of tumor suppressor proteins remains less explored. To address this, Wei et al. (2022) demonstrated a novel TF-DUBTAC **20** platform [52]. It comprises a DNA oligonucleotide linked to the covalent ligand of deubiquitinase OTUB1 (Figure 17). A series of TF-DUBTACs **20**, including FOXO-DUBTAC, p53-DUBTAC, and IRF-DUBTAC, have been developed via a copper-free strain-promoted azide-alkyne cycloaddition (SPAAC) reaction. These compounds stabilized cellular FOXO3A, p53, and IRF3 in an OTUB1-dependent manner. TF-DUBTACs suppressed tumorigenesis via the selective stabilization of tumor suppressor transcription factors.

#### 2.4.3. Nanobody-Based Degrader

Many undruggable proteins have surfaces to which other large biomolecules can bind and act as ligands for TPD. Antibodies display exceptional specificity and significant affinity for their antigens by exploiting such protein surfaces. Nanobodies, fragments of single variable heavy chain domains, retain antigen specificity and have been employed as POI ligands in PROTAC synthesis. In this context, the yeast surface display platform, a cost-effective method, was developed to generate nanobody ligands for proteins [53]. Hemoglobin disorders are treated by the reactivation of fetal hemoglobin (HbF, α_2_γ_2_) [54]. The low survival rates of patients with sickle cell disease are because of the high-level production of HbF [55]. BCL11A subjugates HbF expression, thereby switching from fetal to adult hemoglobin expression during erythropoiesis [56]. Dassama et al. (2022) developed a cell-permeant nanobody fused to a miniature protein and an E3 adaptor to form a degrader **21** (Figure 18) [57]. This nanobody degrader **21** efficiently depleted cellular BCL11A in differentiated primary erythroid precursor cells. Furthermore, it significantly induced fetal hemoglobin expression. This strategy has emerged to compliment traditionally “undruggable” targeted degradation.

### 2.5. PROTAC-Based Approaches

#### 2.5.1. BacPROTACs

ClpC: ClpP (ClpCP) protease is a key quasi-proteasomal particle crucial for microbial protein quality control, stress tolerance, and pathogenicity. ClpCP was also observed in Gram-positive bacteria and mycobacteria. Similar to eukaryotes, only a few bacterial species undergo TPD. PROTAC technology is primarily focused on the ubiquitin tagging of eukaryotes and needs to be extended to bacterial degradation. Phosphorylated arginine residues (pArg) function as a degradation signal, which is recognized by ClpCP in the form of a simple phosphate group attached to the arginine residue of the target protein [58]. Clausen et al. (2022) developed a novel small-molecule degrader, the BacPROTAC **22**. These pArg-containing BacPROTACs **22** recruit POI to the ClpC domain, resulting in their degradation by the ClpCP protease (Figure 19) [59]. These small-molecule adapters redirect ClpCP protease for specifically targeting neo-substrates. Moreover, degradation and drug susceptibility assays using this method were performed in mycobacteria, revealing in vivo activity by selectively targeting endogenous proteins through fusion with an established degron **23**.

The significant benefits of BacPROTACs include (a) potent in vivo activity, (b) selective and efficient degradation of POI in bacteria, (c) cell permeability, (d) wide applicability to bacterial proteins owing to the efficacy of reprogramming ClpCP protease, and (e) the feasibility of repurposing various protein ligands. BacPROTACs have two major limitations. First, the in vivo degradation efficacy is lower than that in vitro. Second, when deregulating the ClpC1P1P2 protease, cyclomarin (Cym)-based degraders might interfere with the cellular levels of unrelated proteins.

#### 2.5.2. DUB-Based Degrader

Deubiquitinating enzymes (DUBs) have emerged as potential targets for treating various human cancers and other diseases. Steinebach et al. (2022) reported the first DUB degrader **24** and revealed a new class of drug targets for TPD [60]. DUB-degraders are PROTACs targeting a ubiquitin-specific protease 7. The design of the degrader adopted glutarimide-based CRBN binder and USP7 inhibitors, which were connected by variable linkers (Figure 20). These small-molecule degraders **24** efficiently induce apoptosis in USP7-dependent cancer cells.

#### 2.5.3. HEMTACs

Heat shock protein 90 (HSP90), an ATP-dependent molecular chaperone, is widely expressed in all eukaryotic cells and displays tumor-selective pharmacokinetics [61,62,63]. HSP90 plays an important role in regulating cytoprotection, cell cycle progression, and cell signaling via its increased production against stress response. Traditional E3 ligase VHL- and CRBN-based small-molecule PROTAC degraders cause off-target effects and resistance mechanism in cells [64,65,66]. To address this, Li et al. (2023) developed heat shock protein 90-mediated targeting chimeras **25** (HEMTACs) for intracellular protein degradation [67]. HSP90 binds to the target protein and facilitates the downregulation of proteins. HEMTACs **25** comprise a POI binding ligand, linker, and HSP90-binding ligand (Figure 21). Cyclin-dependent kinases regulate eukaryotic cell division, proliferation, and gene expression [68]. CDK4/6 is usually overexpressed and hyperactivated in cancer cells and plays a key role in initiating the cell cycle [69,70,71]. By adopting HEMTACs, CDK4/6 was successfully degraded, thus expanding the potential for TPD.

## 3. Conclusions

Drug discovery and development have witnessed rapidly successful protein-degradation technologies in the form of PROTACs and several newly emergent modalities. This is evident from the initiation of a clinical trial for >10 PROTAC molecules. The journey of PROTACs from concept-to-clinic took approximately two decades. Despite the remarkable progress achieved, certain inherent limitations remain to be addressed. These involve issues associated with selectivity, permeability, role of linkers, and druggability. Several new modalities have been developed to address these challenges. However, these emerging modalities are still immature and require further elucidation for obtaining successful PROTAC-based drug candidates. We hypothesize that several PROTAC molecules will be accessible to patients exhibiting various conditions, including cancer, neurodegenerative disorders, inflammation, and viral infections, in the future.

## Figures and Tables

**Figure 1 ijms-24-05190-f001:**
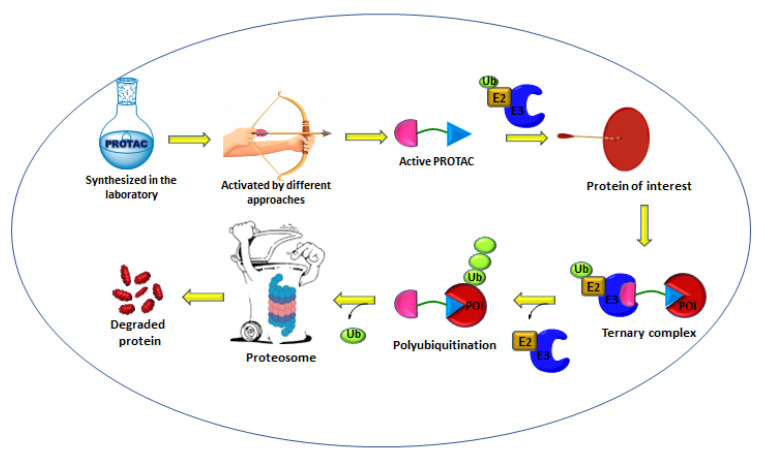
Schematic of the mechanism of proteolysis-targeting chimeras (PROTACs).

**Figure 2 ijms-24-05190-f002:**
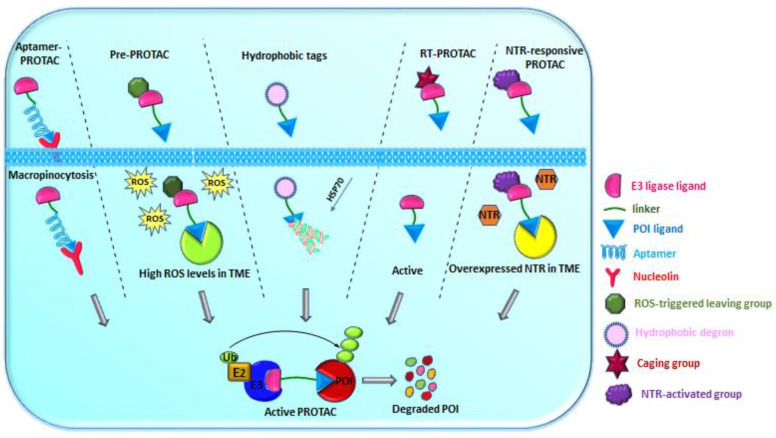
Mode of action—Aptamer–PROTACs: upon recognition by the cell membrane receptor (nucleolin), aptamer–PROTACs enter the cells, and endocytosis followed by linker cleavage result in the liberation of active PROTAC; Pre-PROTACs: inserting reactive oxygen species (ROS)-triggered leaving group to the parent PROTAC results in the release of the active PROTAC in the presence of ROS of the tumor microenvironment (TME); Hydrophobic tags: hydrophobic degron mimics the exposed hydrophobic region of misfolded proteins and is recognized by the heat shock protein for proteasomal degradation; RT-PROTACs: radiotherapy-triggered PROTAC prodrug incorporated with phenyl azide caging group releases the active PROTAC molecule upon X-ray irradiation; NTR-responsive PROTACs: caging of the nitroreductase (NTR)-responsive group releases the active PROTAC in the presence of hypoxia microenvironment.

**Figure 3 ijms-24-05190-f003:**
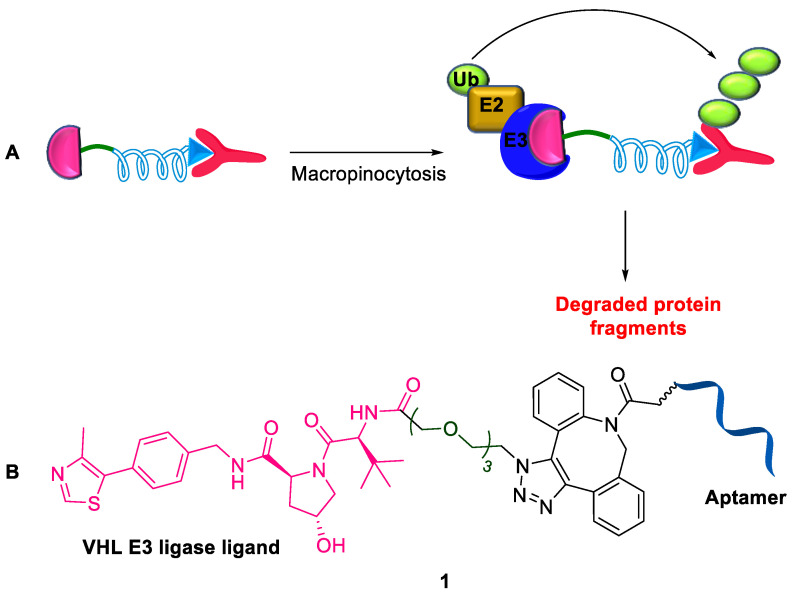
Aptamer-constructed PROTAC: (**A**) mode of action; (**B**) chemical structure.

**Figure 4 ijms-24-05190-f004:**
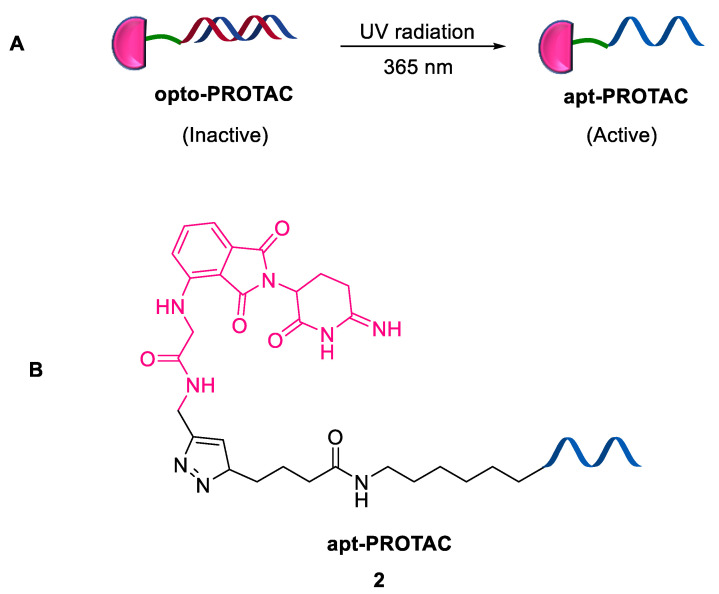
(**A**) Mode of action in opto-PROTAC; (**B**) chemical structure of apt-PROTAC.

**Figure 5 ijms-24-05190-f005:**
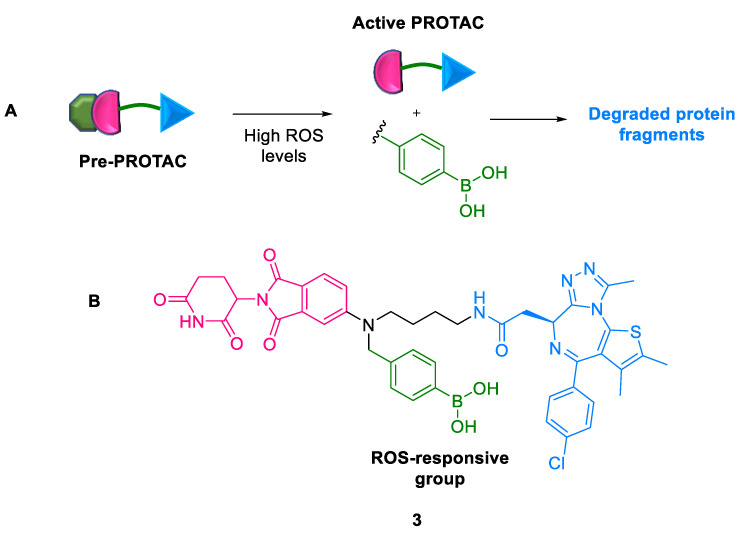
Pre-PROTAC: (**A**) mode of action; (**B**) chemical structure.

**Figure 6 ijms-24-05190-f006:**
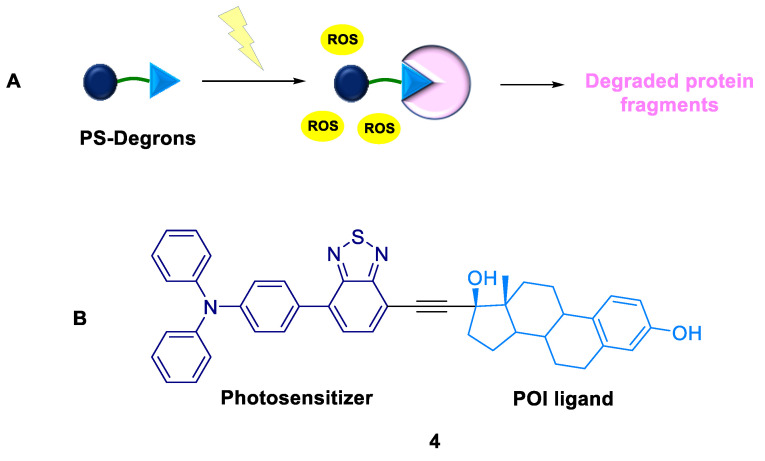
PS-Degrons: (**A**) mode of action; (**B**) chemical structure.

**Figure 7 ijms-24-05190-f007:**
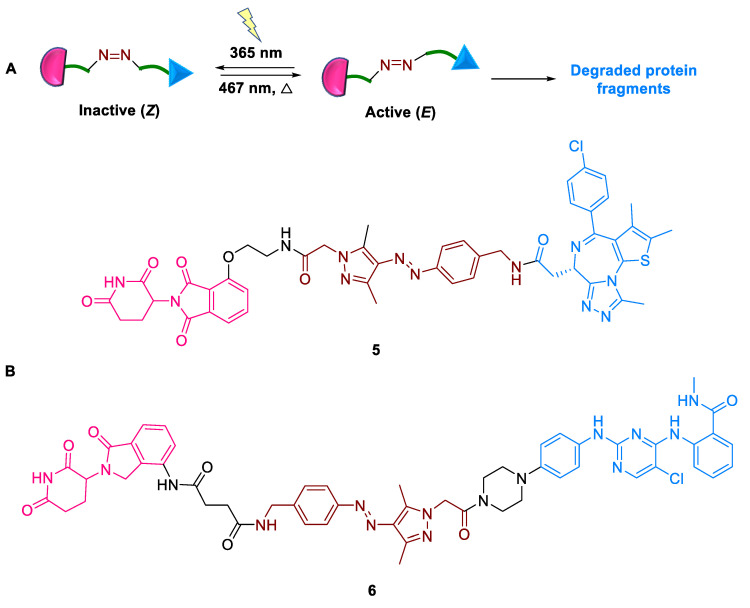
AP-PROTAC: (**A**) mode of action; (**B**) chemical structure.

**Figure 8 ijms-24-05190-f008:**
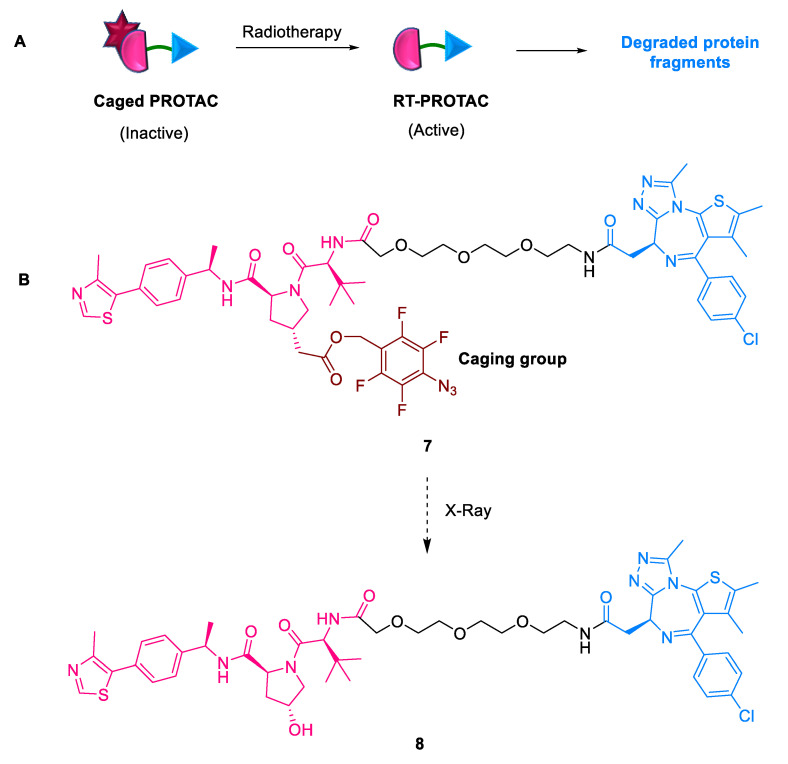
RT-PROTAC: (**A**) mode of action; (**B**) chemical structure.

**Figure 9 ijms-24-05190-f009:**
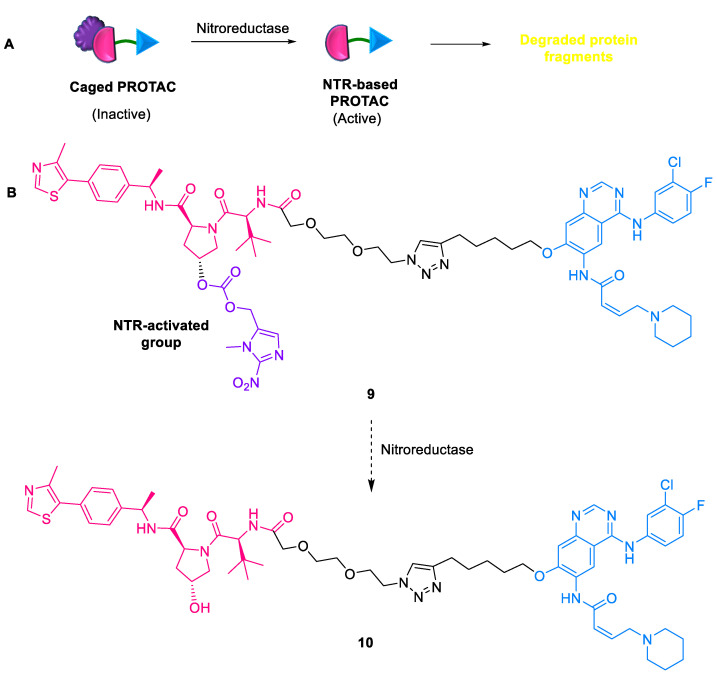
NTR-based PROTAC: (**A**) mode of action; (**B**) chemical structure.

**Figure 10 ijms-24-05190-f010:**
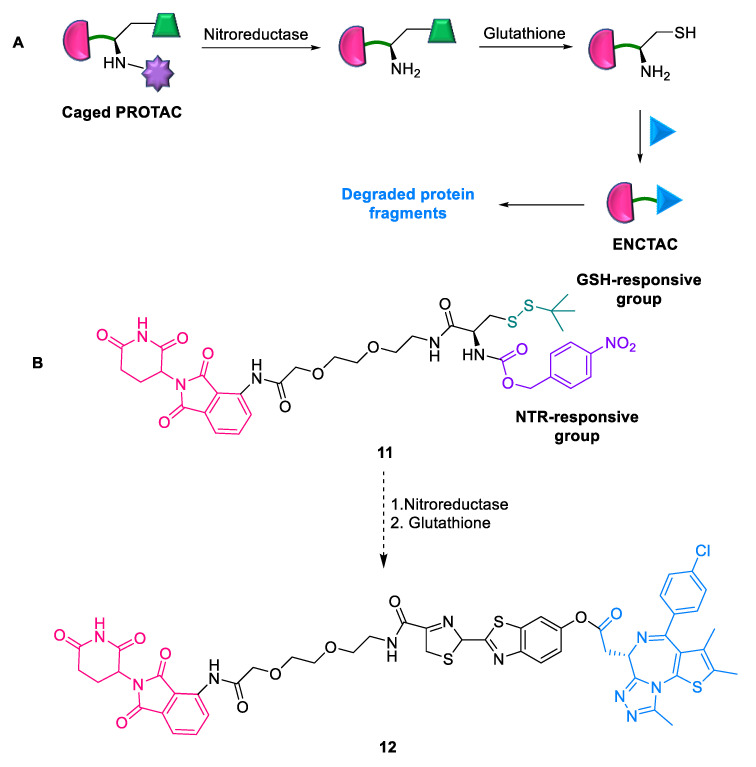
ENCTAC: (**A**) mode of action; (**B**) chemical structure.

**Figure 11 ijms-24-05190-f011:**
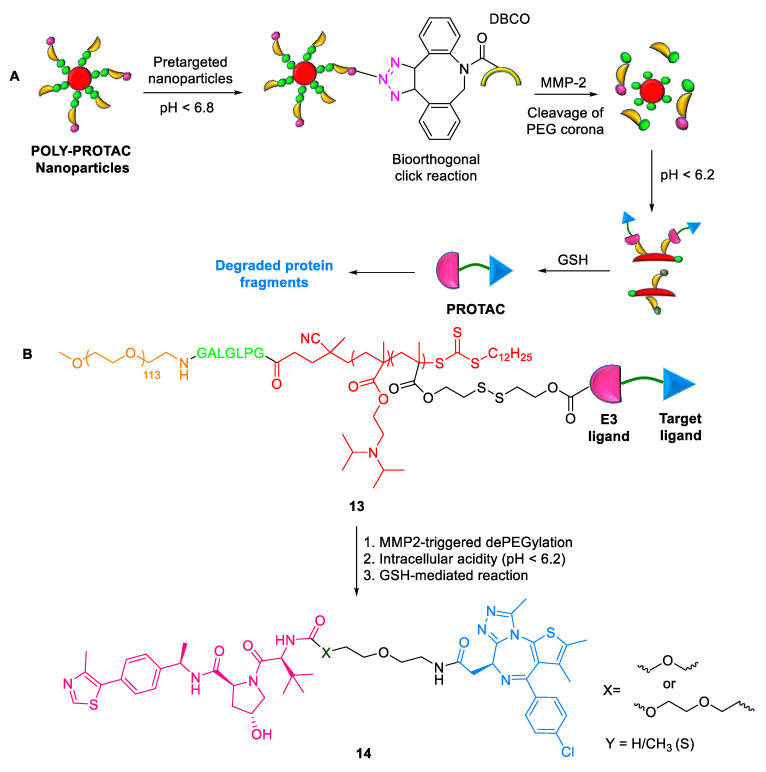
Poly-PROTAC: (**A**) mode of action; (**B**) chemical structure.

**Figure 12 ijms-24-05190-f012:**
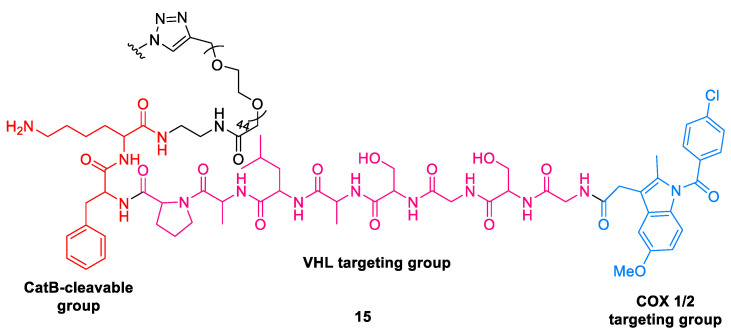
Chemical structure of smart nano-PROTAC.

**Figure 13 ijms-24-05190-f013:**
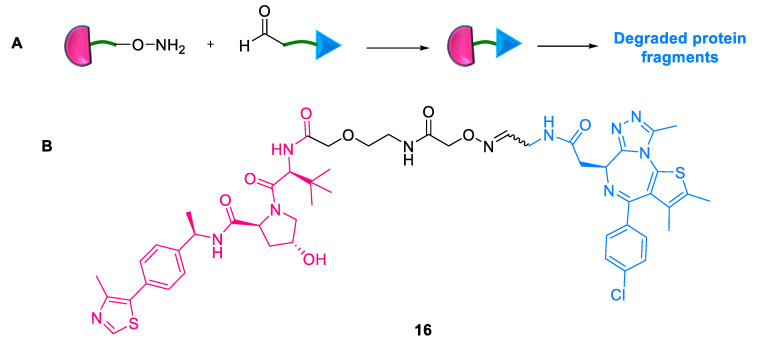
Split-PROTAC: (**A**) mode of action; (**B**) Chemical structure.

**Figure 14 ijms-24-05190-f014:**
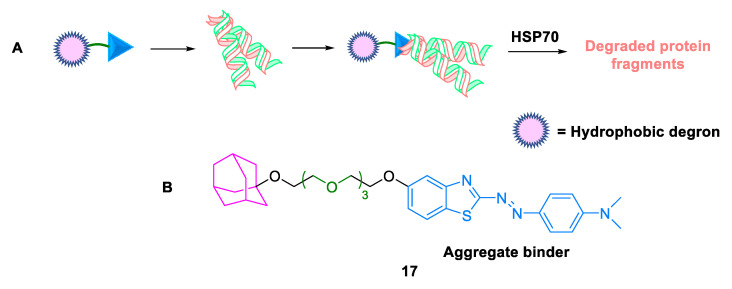
Hydrophobic tag: (**A**) mode of action; (**B**) chemical structure.

**Figure 15 ijms-24-05190-f015:**
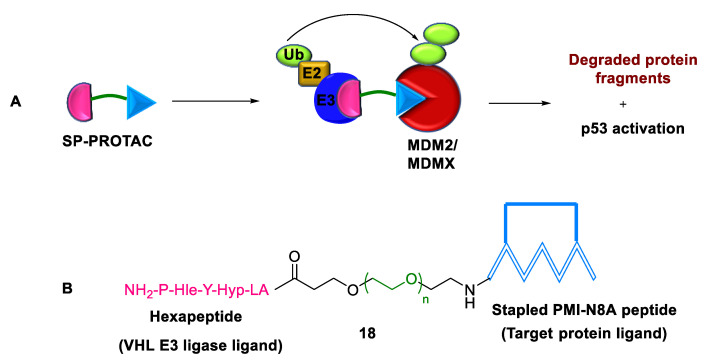
Stapled PROTAC: (**A**) mode of action; (**B**) chemical structure.

**Figure 16 ijms-24-05190-f016:**
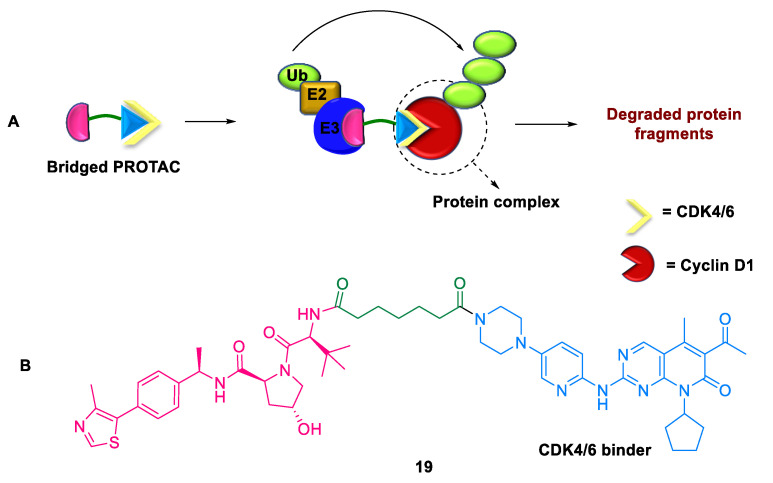
Bridged PROTAC: (**A**) mode of action; (**B**) chemical structure.

**Figure 17 ijms-24-05190-f017:**
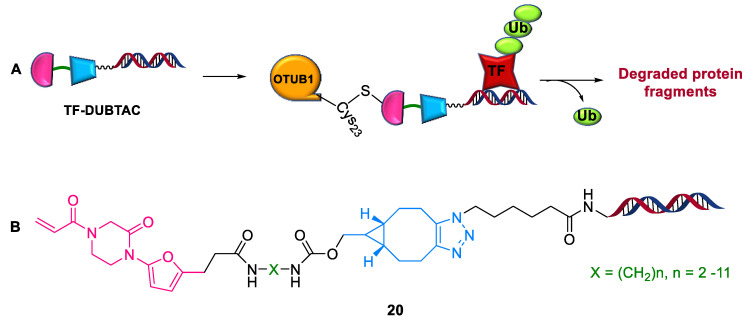
TF–DUBTAC: (**A**) mode of action; (**B**) chemical structure.

**Figure 18 ijms-24-05190-f018:**
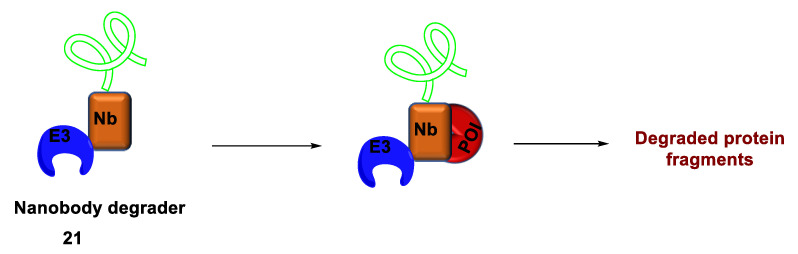
Mode of action by nanobody-based degrader.

**Figure 19 ijms-24-05190-f019:**
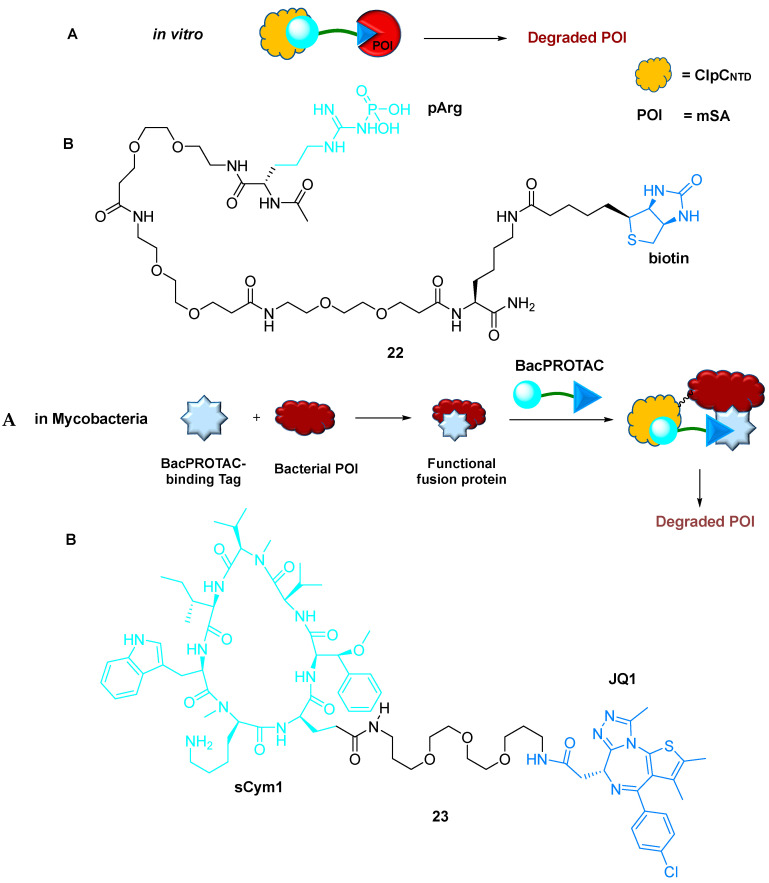
BacPROTAC: (**A**) mode of action; (**B**) chemical structure.

**Figure 20 ijms-24-05190-f020:**
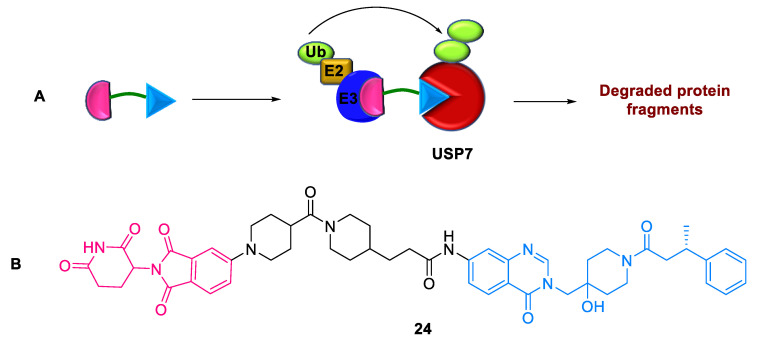
DUB–based degrader: (**A**) mode of action; (**B**) chemical structure.

**Figure 21 ijms-24-05190-f021:**
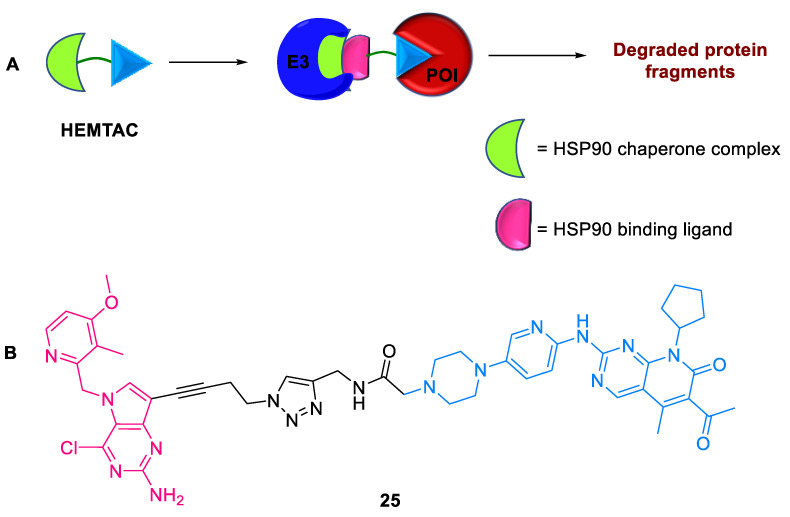
HEMTAC: (**A**) mode of action; (**B**) chemical structure.

## Data Availability

Not applicable.

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
