# Peer review of "Emerging Strategies in Proteolysis-Targeting Chimeras (PROTACs): Highlights from 2022"

_ijms, 2023, doi:10.3390/ijms24065190_

Round 1

Reviewer 1 Report

This review listed a panel of PROTACs based on the categorization of selectivity and controllability, linker flexibility, cell permeability, and druggability. Overall, the review was poorly written, with a lot of PORTACs listed there, but no insights or summary from the authors. It’s quite painful to read through the whole manuscript.

There are also some specific comments listed below:

1.       There are some sentences with conflicts. Authors claimed PROTACs are limited to acquired resistance in line 16, but “PROTACs have received considerable attention since they target the undruggable and overcome drug resistance” in line 47-48.

2.        Line 406-412, a DUBTAC generally means a deubiquitinase is recruited by small molecules to remove ubiquitin from substrates, not a PROTAC degrading a deubiquitinase can be addressed as DUBTAC.

Author Response

Clarification to comments

Reviewer 1

This review listed a panel of PROTACs based on the categorization of selectivity and controllability, linker flexibility, cell permeability, and druggability. Overall, the review was poorly written, with a lot of PORTACs listed there, but no insights or summary from the authors. It’s quite painful to read through the whole manuscript.

There are also some specific comments listed below:

1.   There are some sentences with conflicts. Authors claimed PROTACs are limited to acquired resistance in line 16, but “PROTACs have received considerable attention since they target the undruggable and overcome drug resistance” in line 47-48.

A.   The sentences are modified as per reviewer’s suggestion.

2.   Line 406-412, a DUBTAC generally means a deubiquitinase is recruited by small molecules to remove ubiquitin from substrates, not a PROTAC degrading a deubiquitinase can be addressed as DUBTAC.

A.   “DUBTACs are PROTACs based on a ubiquitin-specific protease 7 inhibitor scaffold.” This is specified in the text.

Reviewer 2

This review discusses Proteolysis-Targeting chimeras (PROTACs) and their therapeutic potential in varieties of diseases. The authors have done an excellent job of discussing all the strategies, advantages, challenges, shortcomings, and advances in PROTAC technology. This review is very field-specific. However, it is written in a such clear way that it will give a very clear picture to the people who want to know about PROTAC technology and its uses. This review is written in a well-organized way with self-explanatory figures. I recommend some minor corrections that can be made to improve the quality of this review.

1. Figure 1 was made in a very creative way, which is commendable. However, it is a bit difficult to understand the meaning of the arrow here. Does that mean that it is directed to make active PROTAC? I think the figure is clear enough without the first two icons unless you put a few words below the icon to explain it. The protein of interest icon can be all red with POI written in the bracket since the POI just below is a red icon.

A. Figure 1 is modified with some added text below the icons. Also, the protein of interest icon is modified as per reviewer’s suggestion.

2. Figures 8, 9, 10, 11, and 13 should have E3 ligand and Protein of Interest (POI) structure marked. Other figures in the paper have color-coded schematics, which makes it easy to locate the respective structure, but it is missing for the above-mentioned figures.

A. Figures 8, 9, 10, 11, and 13 are modified as per reviewer’s suggestion.

3. Line 53: what is a rule of five (bo5)? This point can be conveyed without using this arcane term.

A. The sentence is now modified as “Lipinski’s rule of five.”

4. Figure 2 should include a full form of NTR and TME either in the figure or in the legend.

A. Figure 2 is now modified as per reviewer’s suggestion.

5. tumor and tumour both are used throughout the paper. The authors should stick to either American or British English.

A. The word “tumor” is now used in the whole text.

6There is a duplication of a sentence in lines 168-170.

A. The sentence is modified as per reviewer’s suggestion.

7. There is a repetition of the exact same sentence in the abstract (lines 22-24) and conclusion (lines 440-442). The sentence in the conclusion should be rewritten.

A. Abstract and conclusion is now modified as per reviewer’s suggestion.

8. Formatting can be done better e.g. spacing issues in lines 133, 338,439, etc., in vivo should be in italics in lines 395,397,401 (in vitro too).

A. Formatting and italics is now done in the text.

Reviewer 3

The manuscript title: “Emerging strategies in proteolysis targeting chimeras (PROTACs): Highlights of the year 2022”.

Proteolysis targeting chimera (PROTAC) is an emerging protein degradation strategy, which shows excellent results in preclinical and clinical settings targeting undruggable proteins. However, the potential systemic toxicity of PROTACs caused by undesired off-target protein degradation may still limit the application of PROTACs in clinical practice. In the current manuscript, authors comprehensively reviewed different strategies developed in 2022 to increase target specificity, selectivity, cell permeability by conjugating to Aptamer, or by conjugating a reactive oxygen species (ROS)-triggered leaving group, PS-Degrons, AP-PROTACs, RT-PROTCs, NTR-based PROTACs, enzyme-derived clicking PROTACs (ENCTACs), polymeric PROTAC etc. It is well organized & written manuscript and should be suitable for publication in IJMS in its current forms.

General suggestions

Authors may consider these two general comments before publications.

·    1.  Abstract is too general; it should read the contents covered in the manuscript. 

A.   Abstract is now modified as per reviewer’s suggestion.

2.   Authors should mention the compounds numbers in the text, so that it is easy to follow the readers.

A.   Compound numbers is now added in the text.

3.   Page 8 line 179 needs to be corrected “PHOtochemically Targeting Chimeras (PHOTACs) are a small group of photoswitchable”

A.   Line 179 is now modified as per reviewer’s suggestion.

Reviewer 2 Report

This review discusses Proteolysis-Targeting chimeras (PROTACs) and their therapeutic potential in varieties of diseases. The authors have done an excellent job of discussing all the strategies, advantages, challenges, shortcomings, and advances in PROTAC technology. This review is very field-specific. However, it is written in a such clear way that it will give a very clear picture to the people who want to know about PROTAC technology and its uses. This review is written in a well-organized way with self-explanatory figures. I recommend some minor corrections that can be made to improve the quality of this review.

-         Figure 1 was made in a very creative way, which is commendable. However, it is a bit difficult to understand the meaning of the arrow here. Does that mean that it is directed to make active PROTAC? I think the figure is clear enough without the first two icons unless you put a few words below the icon to explain it. The protein of interest icon can be all red with POI written in the bracket since the POI just below is a red icon.

-         Figures 8, 9, 10, 11, and 13 should have E3 ligand and Protein of Interest (POI) structure marked. Other figures in the paper have color-coded schematics, which makes it easy to locate the respective structure, but it is missing for the above-mentioned figures.

-         Line 53: what is a rule of five (bo5)? This point can be conveyed without using this arcane term.

-         Figure 2 should include a full form of NTR and TME either in the figure or in the legend.

-         tumor and tumour both are used throughout the paper. The authors should stick to either American or British English.

-         There is a duplication of a sentence in lines 168-170.

-         There is a repetition of the exact same sentence in the abstract (lines 22-24) and conclusion (lines 440-442). The sentence in the conclusion should be rewritten.

-         Formatting can be done better e.g. spacing issues in lines 133, 338,439, etc., in vivo should be in italics in lines 395,397,401 (in vitro too).

Author Response

(The authors gave the same response as above.)

Reviewer 3 Report

Reviewers’ comments for the Manuscript ID: ijms-2210045

The manuscript title:Emerging strategies in proteolysis targeting chimeras (PROTACs): Highlights of the year 2022”.

Proteolysis targeting chimera (PROTAC) is an emerging protein degradation strategy, which shows excellent results in preclinical and clinical settings targeting undruggable proteins. However, the potential systemic toxicity of PROTACs caused by undesired off-target protein degradation may still limit the application of PROTACs in clinical practice. In the current manuscript, authors comprehensively reviewed different strategies developed in 2022 to increase target specificity, selectivity, cell permeability by conjugating to Aptamer, or by conjugating a reactive oxygen species (ROS)-triggered leaving group, PS-Degrons, AP-PROTACs, RT-PROTCs, NTR-based PROTACs, enzyme-derived clicking PROTACs (ENCTACs), polymeric PROTAC etc. It is well organized & written manuscript and should be suitable for publication in IJMS in its current forms.

General suggestions

Authors may consider these two general comments before publications.

·       Abstract is too general; it should read the contents covered in the manuscript.  

  • Authors should mention the compounds numbers in the text, so that it is easy to follow the readers.
  • Page 8 line 179 needs to be corrected “PHOtochemically Targeting Chimeras (PHOTACs) are a small group of photoswitchable”

Author Response

(The authors gave the same response as above.)

Round 2

Reviewer 1 Report

It seems that my review opinions were ignored by the authors in the revised manuscript. Again, the USP7 targeting PROTAC should not be named as DUBTAC (from line 402 to 410), I carefully checked the original paper, and they didn’t state this PROTAC as DUBTAC.